# Regulation of Fructose Metabolism in Nonalcoholic Fatty Liver Disease

**DOI:** 10.3390/biom14070845

**Published:** 2024-07-13

**Authors:** Mareca Lodge, Rachel Dykes, Arion Kennedy

**Affiliations:** Department of Molecular and Structural Biochemistry, North Carolina State University, 128 Polk Hall Campus, Box 7622, Raleigh, NC 27695, USA

**Keywords:** fructose, liver, adipose tissue, intestine, immune cells, metabolic disease

## Abstract

Elevations in fructose consumption have been reported to contribute significantly to an increased incidence of obesity and metabolic diseases in industrial countries. Mechanistically, a high fructose intake leads to the dysregulation of glucose, triglyceride, and cholesterol metabolism in the liver, and causes elevations in inflammation and drives the progression of nonalcoholic fatty liver disease (NAFLD). A high fructose consumption is considered to be toxic to the body, and there are ongoing measures to develop pharmaceutical therapies targeting fructose metabolism. Although a large amount of work has summarized the effects fructose exposure within the intestine, liver, and kidney, there remains a gap in our knowledge regarding how fructose both indirectly and directly influences immune cell recruitment, activation, and function in metabolic tissues, which are essential to tissue and systemic inflammation. The most recent literature demonstrates that direct fructose exposure regulates oxidative metabolism in macrophages, leading to inflammation. The present review highlights (1) the mechanisms by which fructose metabolism impacts crosstalk between tissues, nonparenchymal cells, microbes, and immune cells; (2) the direct impact of fructose on immune cell metabolism and function; and (3) therapeutic targets of fructose metabolism to treat NAFLD. In addition, the review highlights how fructose disrupts liver tissue homeostasis and identifies new therapeutic targets for treating NAFLD and obesity.

## 1. Introduction

Worldwide, approximately 46.9 cases of nonalcoholic fatty liver disease (NAFLD) occur among 1000 persons per year [1]. In the U.S., 47.8% of the population is diagnosed with NAFLD, with an average of 30% developing nonalcoholic steatohepatitis (NASH) [1,2,3,4,5]. NAFLD encompasses many pathologies that consist of hepatic lipid accumulation, known as hepatic steatosis, independent of alcohol consumption or other secondary causes of fatty lipid accumulation in the liver [6,7,8,9]. Hepatic steatosis can progress to NASH, defined by inflammation and histological evidence of liver injury [10,11]. NASH is the third most common reason for liver transplants and 5–25% of patients will progress to cirrhosis, with 13% progressing to hepatocellular carcinoma, one of the most common causes of cancer-related deaths [10,12,13,14,15,16,17]. 

The incidence of NAFLD is increasing each year and is related to several metabolic and lifestyle factors. The increase in individuals with NAFLD is partly due to a rise in the number of patients with Type 2 diabetes mellitus (T2DM), with more than 50–70% of those with T2DM diagnosed with NAFLD [18,19,20,21]. NAFLD is also strongly associated with metabolic syndrome (MetS), as obesity, insulin resistance, and hyperlipidemia are common risk factors [22,23,24,25,26,27,28]. Resmetirom is the only therapy for NASH approved by the U.S. Food and Drug Administration (FDA), but it remains to be seen if this therapy will lead to long-term complications or side effects [29]. Therefore, it is necessary to identify new molecular mechanisms driving the progression of NAFLD to develop effective therapies with minimal complications. 

Diet plays a key role in the development of NAFLD. The typical Western diet, characterized by excess fat and carbohydrates, mainly from sugars such as fructose, has been shown to drive obesity and MetS [30,31,32]. Fructose consumption has increased over the past 40 years among adults and children [33,34,35]. Dietary fructose is often accompanied by other sugars which are consumed in excess, such as glucose, sucrose, or high-fructose corn syrup (HFCS) [36]. Sucrose, commonly known as table sugar, consists of a 50:50 ratio of fructose to glucose, while HFCS consists of ~50–55% fructose and the remaining glucose. These sugars are present in yogurts, fast food, and highly consumed sugar-sweetened beverages. While fructose is often consumed in the form of sucrose, both sucrose and HFCS can elevate levels of triglycerides, free fatty acids, and insulin release after consumption [37,38,39]. 

Fructose metabolism is dependent on the breakdown of adenosine triphosphate (ATP), generating adenosine monophosphate (AMP) which leads to increased uric acid production [40,41]. Zheng et al. recently reported that elevated fructose-to-vitamin C ratios are positively associated with increased serum uric acid levels among African American men [42]. Hyperuricemia and high consumption of fructose also are strongly correlated with the incidence of NAFLD and cardiovascular disease [43,44,45,46,47]. Elevated serum levels of uric acid are strong predictors of NAFLD among obese adults [48]. Due to the harmful effects of increased consumption of sucrose or HFCS on human health, it is imperative to fully distinguish the unique effects of fructose on the development of metabolic disease.

The current dogma regarding fructose metabolism and metabolic disease suggests hepatocytes and enterocytes metabolize fructose, altering glucose and lipid metabolism. In addition, intestinal microbiomes exposed to high concentrations of fructose produce metabolites that contribute to progression of NAFLD. However, there are a number of other cell types exposed to high concentrations of fructose, including immune cells, that interact with parenchymal cells in metabolic tissues to control homeostasis. Myeloid cells and lymphocytes are specifically known to contribute to inflammation and fibrosis in metabolic tissues. However, a knowledge gap remains in our understanding of the role of fructose metabolism in immune cells when exposed to an excess of fructose and the pathogenesis of metabolic diseases, such as NAFLD. The present review highlights the current literature regarding how fructose regulates inflammation in metabolic tissues, the implications of fructose metabolism on immune cell phenotypes, and therapeutic targets of fructose metabolism.

## 2. Fructose Metabolism and Metabolic Tissues

### 2.1. Overview of Fructose Metabolism

In healthy adults, normal resting levels of fructose within peripheral blood are 0.04 mM. However, following fructose consumption, peripheral blood levels can increase to 0.4 mM [49,50]. In the small intestine, polysaccharides are broken down into monomeric forms of glucose, fructose, and galactose. Fructose is taken up into the intestine by glucose transporters (GLUT) 2 and 5. The process of fructose transport is passive and insulin-independent, unlike glucose. The knockout of GLUT5 in mouse models leads to a 75% reduction in fructose absorption in the intestine, highlighting its importance in fructose metabolism [51]. GLUT5 is developmentally regulated, with GLUT5 gene expression significantly lower in fetal intestines than in adults [52]. The consumption of fructose by toddlers leads to cases of malabsorption and early-onset weight gain [53,54]. GLUT5 also is expressed in tissues such as brain, kidney, and skeletal muscle and certain types of cancers [55,56,57,58]. Human monocytes express GLUT5, which increases during macrophage differentiation and foam cell development [59,60]. It is possible that with increased levels of fructose consumption, traditionally non-metabolizing fructose cells become subject to fructose-mediated cell plasticity.

Once taken up into the cells, ketohexokinase C (KHK) phosphorylates fructose. The KHK gene encodes two isotopes, KHKC and KHKA. KHKC is primarily expressed in humans and mice in the liver, kidney, and jejunum, with low expression levels in the pancreas and white adipose tissue [61]. KHKA is ubiquitously expressed and has a slower enzymatic rate compared with KHKC, with the Km values being 0.8 mmol/L and 7 mmol/L for fructose, respectively. Although both isoforms can phosphorylate fructose, the majority of fructose uptake and metabolism is mediated by KHKC in the intestine and liver [62]. Pharmacological inhibition and global knockdown of both isoforms of KHK reduce fructose caloric intake, leading to reductions in body weight, adipose tissue expansion, hepatic lipid accumulation, liver injury, and fibrosis [63,64]. Knockdown of KHK in epithelial cells of the small and large intestines in mice leads to reductions in fructose caloric intake, but does not affect chronic fructose-induced lipid accumulation in adipose tissues or the liver [62]. In contrast, hepatocytes were found to be the primary cells responsible for driving fructose-induced MetS. Interestingly, during the development of hepatocellular carcinoma and breast cancer, there is a switch from KHKC to KHKA activity, leading to the phosphorylation of phosphoribosyl pyrophosphate synthetase 1, driving nucleotide synthesis [65,66]. 

Transcriptional regulation of KHK modulates fructose metabolism and mitigates its effects on intestinal and hepatic lipogenesis and inflammation. Carbohydrate regulatory element binding protein (ChREBP) is the master transcription factor of KHK. Recently, hypoxia inducible-factors alpha-2a (HIF-2α) was discovered to inhibit fructose metabolism by reducing KHK isoforms and aldolase B expression in hepatocytes. In addition, peroxisomal biogenesis factor 2 (PEX2), a peroxisomal membrane protein necessary for peroxisome biogenesis, was also found to control KHK expression in the liver [67]. Fructose metabolism also involves the enzyme hexokinase (HK) [61]. HK phosphorylates fructose to form fructose-6-phosphate, using ATP as a phosphate donor. The HK family is a group of enzymes, including HK I, II, III, and IV which are expressed in liver, while HK I and II are present in adipose tissue [61]. The interplay between fructose and HK is complex, and alterations in fructose metabolism can have implications for metabolic health, including conditions such as NAFLD and MetS. 

Unlike HK, the metabolism of fructose by KHK has no regulating feedback loop, thus allowing for the rapid conversion of fructose. Within the intestine, fructose is converted to glucose, glycerate, and tricarboxylic acid cycle (TCA) metabolites, which are then released into the blood. Although the intestine can metabolize low levels of fructose, it can become overly saturated, and excess fructose is transported through the portal vein and metabolized in the liver [68]. GLUT8 is responsible for fructose uptake in hepatocytes and stellate cells, de novo lipogenesis (DNL), and hepatic inflammation [69,70]. Knockdown of GLUT8 in hepatocytes attenuates DNL and inflammation while increasing fatty acid oxidation, lipolysis, and inflammation in stellate cells. GLUT2 also is expressed on hepatocytes and is capable of fructose uptake with a Km ~67 mM [71]. Following entry into a cell, fructose is further metabolized by KHK and aldolase B [72]. KHK phosphorylates fructose to fructose-1-phosphate. Fructose-1-phosphate is then converted to dihydroxyacetone phosphate and glyceraldehyde by aldolase B, which is further converted to glyceraldehyde-3-phosphate [73]. Triose kinase expressed in hepatocytes is the enzyme responsible for converting glyceraldehyde to glyceraldehyde-3-phosphate and directs fructose metabolites into lipogenesis, gluconeogenesis, and triglyceride accumulation in hepatocytes [73]. Although fructose is primarily shuttled through the portal vein, it is projected that 20–30% of ingested fructose may enter systemic circulation, where it can be deposited into other tissues, including the proximal tubes within the kidney [74]. In addition, fructose is also metabolized by immune cell populations in tissues and peripheral circulation [75,76]. Microglia, the resident macrophages in the brain, also express GLUT5 and KHK and metabolize fructose to drive cognitive dysfunction and influence the progression of metabolic diseases [77]. Ultimately, the concentration and frequency of consumption of dietary fructose determines the metabolic fate of these carbohydrates in metabolic tissues [68].

Endogenous fructose production depends on the isomerization of glucose, mannose, or xylose in bacteria and yeast [62,78]. In humans and other mammals, fructose can be endogenously produced by the polyol pathway, which converts glucose into sorbitol using aldose reductase and fructose using sorbitol dehydrogenase [79]. Under hyperglycemic conditions, fructose is generated through the polyol pathway to increase KHKA activity in cancer cells, promoting metastasis [80]. Interestingly, the activation of the polyol pathway in the liver and the intestine are positively associated with complications of diabetes and the progression of NAFLD [81,82,83].

### 2.2. Fructose and Intestine

Under normal intestinal conditions, fructose has the same metabolic fate in the form of sucrose or fructose alone [13,68]. Fructose carbons are rapidly converted to glucose, glycerate, and TCA metabolites within enterocytes, the primary cell type within the intestine. These byproducts are typically shuttled through the portal vein to the liver for further processing and metabolism. In enterocytes, fructose is converted to fructose-1-phosphate by KHK (Figure 1). Elevated intracellular levels of fructose-1-phosphate can be toxic, leading to endoplasmic reticulum stress and inflammation, which is reported to induce barrier deterioration and shorten colon length [84]. Taylor et al. discovered that fructose promotes hypoxic cell survival via extended villi length, partly through fructose-1-phosphate inhibition of the hypoxic adaptation protein pyruvate kinase isozyme M2 (PKM2) [85]. The extension of the length of the villi increased weight gain, nutritional absorption, and fat accumulation. High levels of fructose lead to a misshapen colon and cecum as well as infiltration of inflammatory macrophages, neutrophils, dendritic cells, and natural killer T cells into the lamina propria. These cells express the inflammatory cytokines interleukin 1 beta (IL-1β), interleukin 2 (IL-2), and interleukin 6 (IL-6) within the small intestine of mice [86,87]. In addition, cecal metabolites such as prostaglandin B1 and I2, which are important for regulating blood pressure, inflammation, and fibrosis, are elevated in male mice on a fructose diet [88]. Although fructose metabolism begins within the small intestine at low doses, it can bypass initial metabolism at higher doses and be shuttled through the portal vein for further processing [89]. 

High concentrations of fructose reduce the ratio of *Bacteroidetes/Firmicutes* while increasing the abundance of the bacterial genera *Bifidobacterium*, *Enterococcus*, *Lactobacilus*, *Romboutsia*, and *Turicibater* [86,90]. These microbes metabolize fructose through fructokinase, the phosphoenolpyruvate-dependent sugar phosphotransferase system, and other hexose metabolism pathways [91,92,93,94,95]. Additionally, the pro-inflammatory metabolites cresol, arachidonic acid, stearic acid, palmitic acid, and indole sulfuric acid were demonstrated to be elevated, while anti-inflammatory short chain fatty acids (SCFAs) such as acetic acid, propionic acid, butyric acid, and pentanoic acid were reduced in male mice and humans consuming high-fructose diets [86,96]. Among obese and overweight teenagers, *Eubacterium* and *Streptococcus* were found to be negatively correlated with fructose consumption [97]. The gut bacteria of obese youth also are reported to have a higher capability to ferment fructose compared with microbes in the guts of lean subjects [98]. The abundance of *Barnesiella* is elevated, while the abundance of *Ruminococcus*, *Faecalibacterium*, and *Erysipelatoclostridium* are reduced among healthy adult women consuming a HFCS diet [96]. Fructose malabsorption can occur when dietary concentrations exceed enterocyte absorption capacity [99]. In addition to increased bacterial growth, Kawabata et al. found that a high fructose consumption induced a leaky gut by decreasing protein expression of the tight junction protein occludin in male mice [100]. Ultimately, a leaky gut may lead to the increased permeability of the epithelial cells lining the digestive tract and induce inflammatory cell infiltration. 

### 2.3. Fructose and Liver

Over the last decade, studies have linked monosaccharide fructose to the progression of NAFLD, as fructose consumption has increased from the recommended 5% of daily energy intake to 17% within sugar-sweetened beverages alone, correlating with the rise in obesity-related diseases [101]. As high doses of fructose saturate intestinal metabolism, the remaining fructose is metabolized in the liver (Figure 1). The metabolism of fructose produces triose phosphates that form acetyl-CoA, the essential building block for fatty acid synthesis. Fructose metabolism stimulates DNL through triose phosphate-activated fatty acid synthesis and transcriptional regulation of enzymes involved in DNL. Softic et al. demonstrate that male mice consuming fructose-sweetened water in the presence of a high-fat diet exhibited increased fatty acid synthesis and hepatic insulin resistance compared with glucose-sweetened water in a process dependent on KHK [102]. Fructose supplementation increases the levels of the transcription factors sterol regulatory element binding protein 1 (SREBP-1) and ChREBP, as well as the downstream targets acetyl-CoA carboxylase (ACC), fatty acid synthase (FAS), and the fatty acid transporter CD36 [103]. Fructose metabolism also reduces hepatic ATP levels and the activation of AMP-activated protein kinase (AMPK), which impairs insulin signaling and increases DNL [104,105,106]. 

Multiple studies in male mice and rats have demonstrated that a high-fat fructose diet leads to increased infiltration of macrophages within hepatic tissues [107,108,109]. Furthermore, emerging evidence indicates the recruitment of myeloid cells from the bone marrow and mature macrophages originating from surrounding tissues, such as peritoneal and intestinal macrophages [110]. The emergence of single-cell RNA sequencing has allowed for the identification of residential and recruited macrophage subsets based on gene signatures [111]. In addition, Kupffer cells, residential macrophages of the liver, or recruited macrophages may express a fluid phenotype, displaying pro- and anti-inflammatory properties. External stimuli such as lipopolysaccharide (LPS) or interferon-gamma (IFNγ) activate Toll-like receptors (TLR) of macrophages to secrete cytokines such as IL-6 and tumor necrosis factor alpha (TNFα) to aid in-wound repair through the increased flux of glycolysis [112]. During prolonged inflammation, macrophages can take on an anti-inflammatory phenotype, releasing extracellular matrix proteins to signal wound healing. A chronic fructose diet increases hepatic transitioning monocytes and reduces Kupffer cells in male mice [75]. In vitro studies demonstrate that Kupffer cells metabolize fructose through glycolysis and the pentose phosphate pathway. In addition to increased monocyte infiltration, Kupffer cells express a wound healing and anti-inflammatory phenotype, indicating fructose regulates Kupffer cell phenotype and viability (Figure 2). This prolonged, chronic inflammation can lead to tissue damage and, ultimately, result in the development of NASH. 

### 2.4. Fructose and Adipose Tissue

Adipose tissue is dynamic in cell populations and crosstalk fluidity, thus multiple cell types are exposed to dietary carbohydrates. Fructose also increases adipogenesis [113] (Figure 1). Adipocytes depend on GLUT5 to transport fructose and HK for phosphorylation [114]. Once isomerized to glucose-6-phosphate (G6P), the fate of fructose diverges through glycolysis. Using C^13^-labeled fructose, fructose has been demonstrated to be shuttled into acetyl-CoA and incorporated into the TCA cycle, causing an increase in the release of fatty acid and palmitate [115]. In the presence of glucose, fructose increases the production of nicotinamide adenine dinucleotide phosphate (NADP) and carbon dioxide, which can be utilized for lipid storage. Fructose increases gene expression of peroxisome proliferator-activated receptor γ (PPARγ), CCAAT/enhancer-binding protein alpha (C/EBPα), GLUT4, adiponectin, and adipocyte fatty acid binding protein 2 (AP2) in 3T3-L1 preadipocytes [116]. GLUT5 was determined to be essential for adipocyte differentiation, as knockdown of this transporter reduced the expression of PPARy and AP2, as well as the ability of adipocytes to store lipids. Fructose also reduces beta-oxidation by decreasing PPAR beta and delta, leading to increased inflammation in adipose tissue and the liver [117]. Adipocytes maintain homeostasis by secreting cytokines to signal for immune cell recruitment. Fructose increases the secretion of the pro-inflammatory cytokines interleukin 18 (IL-18), IL-1β, and TNFα in the adipose tissue of male and female rats [118]. Interestingly, the anti-inflammatory proteins interleukin 10 (IL-10) and nuclear receptor factor 2 (Nrf2) also were elevated, potentially to reduce the pro-inflammatory response [118]. Taken together, these studies indicate a direct effect of fructose exposure on energy production and maturation of adipocytes.

### 2.5. Fructose Metabolism Regulation of Immune Cell Function

#### 2.5.1. Myeloid Cells

Many studies have examined fructose metabolism in the parenchymal cells of metabolic tissues, such as the intestine, liver, and adipose tissue. However, macrophages can also metabolize fructose to regulate their function and phenotype (Figure 2) [59,60,77]. Glucose is known as the primary fuel source for pro-inflammatory macrophages, with glycolysis providing energy for the production and secretion of cytokines. Peripheral blood monocytes metabolize fructose and shuttle carbons through oxidative phosphorylation (OXPHOS) compared with the lactate production that occurs with glucose metabolism [76]. TCA metabolites such as succinate, malate, and fumarate were also elevated to a greater extent than glucose, most likely to produce energy and reducing agents to maintain increased OXPHOS. The inflammatory cytokines IL-6, IL-1β, IL-18, and TNFα are detected at higher concentrations in fructose-treated monocytes compared with those treated with glucose [76]. This response was, in part, mediated by upregulated signaling of mTORC1. Glutamine carbons mainly contributed to TCA cycle intermediates and cytokine secretion within fructose treatments. These findings demonstrate that fructose directly regulates inflammation in monocytes.

Although macrophages regulate inflammation and wound repair, the liver relies on other immune cells to patrol the environment for damage and infection. Dendritic cells (DCs) can process and present antigens to surrounding lymphocytes to commence and regulate an adaptive immune response [119]. In human dendritic cells, 15mM fructose treatment induces a chronic inflammatory protein secretion of IL-1β, TNFα, and IL-6 [120]. Fructose-treated DCs stimulate T cell IFNγ secretion more significantly than glucose. This response was initiated and controlled by the secretion of TNFα from fructose exposed DCs. The DCs had elevated extracellular acidification rate (ECAR) levels, indicating a metabolic flux through glycolysis to keep up with the demands of cytokine secretion. This process is most likely regulated by the activation of nuclear factor kappa b subunit (NFκB) via advanced glycation end products (AGE) as these proteins were upregulated with fructose exposure [120]. 

#### 2.5.2. B Cells

B cells play a critical role in the progression of NAFLD via the regulation of T cells and fibrosis [121]. However, specific subsets of B cells are known to drive the progression of NAFLD, while others prevent progression [122,123]. Pathogenic B2 cells produce pro-the inflammatory cytokines TNFα, IL-6, and IL-1β, which attract other immune cells to the liver [124]. Additionally, B cells act as antigen-presenting cells to activate T cells and amplify hepatic inflammation and fibrosis. The antibodies produced by B cells can trigger liver injury and fibrosis in NASH. Using C^13^ labeling nuclear magnetic resonance (NMR), carbohydrate metabolism differs based on the carbon source in B cells isolated from blood [125]. Fructose at 15mM is phosphorylated by HK in B cells (Figure 2). Interestingly, when fructose and glucose are supplied together, HK-mediated metabolism is constrained by glucose affinity, leaving fructose metabolism dependent upon the KHK pathway. This switch in metabolism causes DHAP and lactate buildup from methylglyoxal partitioning. Fructose supplementation reduces B cell lymphopoiesis in bone marrow and splenic marginal zone B cells compared with glucose in male mice [126]. Tan et al. demonstrated that glycolysis and oxidative phosphorylation are the primary metabolic pathways for the generation of B cells [126]. Currently, only one study has demonstrated that a chronic fructose diet decreases hepatic B cells in male mice compared with glucose or control diets [75]. 

## 3. Regulation of Fructose Intake and Metabolism to Prevent Metabolic Disease

### 3.1. Dietary Modifications and Physical Activity

Reducing fructose intake and metabolism can effectively treat obesity and NAFLD in men and women [127]. Patients with NAFLD are informed to limit soda, juice, and sports drinks that are high in sugar, which are associated with higher pro-inflammatory markers, triglyceride/high-density lipoprotein cholesterol ratios, and intrahepatic lipids [128,129,130]. In addition, selecting low-fructose-containing fruits and vegetables and avoiding foods with high concentrations of fructose and HFCS is recommended. Consuming high-fiber foods, such as vegetables, legumes, and whole grains, slows down the absorption of carbohydrates and promotes excretion. Soluble fibers, such as psyllium husk, oats, legumes, and insoluble fibers, reduce absorption of fructose and improve liver enzymes, intrahepatic lipid content, and liver fibrosis [131,132,133,134]. Maintaining a balanced gut microbiota through the consumption of probiotics or prebiotics also may help modulate fructose metabolism and reduce its detrimental effects on the liver. The fructose and water-soluble polymer inulin, fermented in the small intestine and colon to SCFAs, provides an energy source for the growth of protective Bifidobacterium [135,136,137]. Regular exercise improves insulin sensitivity and metabolic health and attenuates the fructose-mediated glycemic response in healthy adults, which can mitigate the negative impact of fructose on liver metabolism [138,139,140,141,142]. While effective, these interventions can be challenging for patients to adhere to over time, requiring other therapies to help reduce the harmful effects of increased fructose consumption. 

### 3.2. Nutritional Supplements

Antioxidants may help to alleviate fructose-induced liver damage, oxidative stress, and inflammation to improve overall health (Figure 3A). Both vitamin E and resveratrol attenuate fructose-induced hepatic steatosis and liver damage by reducing lipid peroxidation and inflammation in mice and rats [143,144,145,146]. Fisetin, a naturally occurring flavonoid, has been reported to reduce fructose-induced liver damage in animal models by suppressing NFkB and inducing the transcription factor Nrf2, which regulates antioxidant enzymes [147]. However, limited studies demonstrate that these antioxidants reduce levels of liver enzymes and steatosis among patients with NAFLD [148,149,150,151]. 

L-carnitine is critical in transporting fatty acids into the mitochondria, the cellular organelles responsible for energy production by facilitating beta-oxidation. Studies exploring the potential benefits of L-carnitine supplementation for NAFLD suggest that L-carnitine supplementation improves the levels of liver enzymes, reduces the accumulation of liver fat, and even ameliorates insulin sensitivity in hepatocyte, animal, and human studies [152]. L-carnitine could enhance the transport of fatty acids into the mitochondria, preventing the excessive accumulation of lipids in liver cells induced by a high fructose intake. Hepatic long-chain acylcarnitines (LACs) with 18 and 20 carbons are increased in fructose-fed livers [75]. Serum LACs are positively correlated with the progression of fibrosis and hepatocellular carcinoma and are elevated with fructose supplementation in male mice [102,153]. LACs are elevated when triglyceride synthesis is inhibited and may act in an autocrine or paracrine mechanism to induce inflammation and cell death in macrophages and hepatocytes. Dietary fructose inhibits fatty acid oxidation in the liver and impairs mitochondrial function, which correlates with elevations in long-chain acylcarnitines [102]. In addition, L-carnitine exhibits antioxidant properties by reducing reactive oxygen species (ROS) and inducing the expression of the antioxidant enzyme superoxide dismutase 2 to reduce oxidative stress and the associated inflammation in fructose-treated hepatocytes [154]. 

### 3.3. Pharmacological Interventions

Targeting fructose metabolism for potential therapeutic interventions in biomolecules involves various strategies (Figure 3B). The design and development of small molecules or compounds that selectively inhibit KHK or HK, which could reduce the metabolism of fructose and its downstream effects on DNL and insulin resistance, are at the forefront of the field. Targeting fructose metabolism with KHK inhibitors has several potential therapeutic benefits, including treating obesity, type 2 diabetes, and fatty liver disease [64,155]. In one study, patients with T2DM treated with KHK inhibitors experienced significant reductions in blood sugar levels. KHK inhibitors also improve insulin sensitivity and reduce inflammation [64]. Bempedoic acid, an FDA-approved drug for hypercholesterolemia, reduces high fat and fructose diet-induced hepatic KHK protein expression, fructose uptake, and hepatic beta-oxidation in female rats [156]. Numerous clinical trials on KHK inhibitors have yielded opposing results. One of the most promising clinical trials was a Phase 2 trial of PF-06835919, a reversible KHKA/C inhibitor developed by Pfizer [157,158]. In this trial with 164 participants, PF-06835919 effectively reduced liver fat and inflammation in patients with NAFLD at 16 weeks. In addition, patients experienced significant reductions in insulin concentrations. However, PF-06835919 at 300 mg or 150 mg did not change fasting plasma cholesterol or low-density lipoprotein cholesterol levels. The KHK inhibitor had minimum safety concerns, with both placebo group and groups receiving the high and low concentration of the inhibitor reporting similar treatment-emergent adverse events ranging from diarrhea to hypoglycemia. 

Other enzymes involved in fructose metabolism, such as aldolase B, triokinase, aldose reductase, and sorbitol dehydrogenase, can be targeted to modulate the overall flux of fructose metabolism and its impact on liver function. While targeting these enzymes with inhibitors holds promise as a potential therapeutic agent for targeting fructose metabolism, there also are limitations and challenges. For example, aldolase B, the enzyme responsible for breaking down fructose-1-phosphate into glyceraldehyde and dihydroxyacetone phosphate, was initially considered a potential therapeutic target for colon cancer as it mediates the metabolic reprogramming necessary for tumor metastasis [159]. However, patients with a mutation in the aldolase B gene developed hereditary fructose intolerance (HFI), leading to liver dysfunction and even failure, making this treatment toxic unless administered alongside a fructose-restricted diet [160]. Ultimately, due to these potential harmful off-target effects, therapeutically targeting aldolase B to regulate fructose metabolism was deemed undesirable. High selectivity is crucial to avoid off-target effects on other enzymes or metabolic pathways. Complete inhibition of these enzymes may disrupt glucose metabolism and have systemic metabolic consequences beyond the liver. Therefore, careful assessment of tissue and cell-specific expression patterns and potential off-target effects is necessary. 

### 3.4. Fructose Transporter Modulators

Fructose transporter inhibitors may selectively target fructose metabolism by reducing fructose absorption and metabolism, improving hepatic insulin sensitivity. Inhibitors specifically targeting GLUT5 or GLUT8 could limit fructose availability for metabolism in the liver. GLUT5 is primarily expressed by intestinal epithelial cells and adipose tissue. However, under NASH conditions GLUT5 gene expression is increased in patients with NASH compared to those with normal and steatotic livers [63,161]. The increased expression may be due to infiltrating peripheral monocytes, which are reported to express GLUT5 [60]. Tripp et al. developed a yeast-based screening system to identify potential GLUT5 inhibitors [162]. The inhibitors N-[4-(methylsulfonyl)-2-nitrophenyl]- 1,3-benzodioxol-5-amine (MSNBA) and (−)-epicatechin-gallate (ECG) selectively inhibited GLUT5 without affecting other glucose transporters. Treatment with these inhibitors reduced the growth of mutant yeast cells in fructose-rich media [162,163]. ECG attenuates high-fat-diet-induced hepatic steatosis, ferroptosis, and oxidative stress [164]. MSNBA prevents the fructose-induced proliferation of colon cancer cells (HT-29) with little impact on the neighboring, healthy intestinal mucosa [57]. ECG, a flavonoid belonging to the group of catechins found in tea, cocoa, and fruits, reduces fructose uptake in Caco-2 cells and high-fat and high-fructose-induced obesity and NAFLD in male mice [165,166]. SW157765 is the only inhibitor reported to bind to GLUT8 and reduces glucose uptake in GLUT8 expressing cancer cells [167]. 

### 3.5. Insulin-Sensitizing Agents

The anti-hyperglycemia drug metformin is reported to reduce levels of liver aspartate aminotransferase (AST) and alanine aminotransferase (ALT) in patients with T2DM and NAFLD [168,169,170,171,172,173,174]. In a study conducted by Spruss et al., mice supplemented with metformin showed significantly lower hepatic steatosis, liver injury, and levels of plasma leptin and retinol-binding protein 4 (RBP-4) [174]. Additionally, metformin reduced the hepatic expression of TNFα [175]. The direct target of metformin remains to be determined. Metformin indirectly activates AMP-activated kinase and reduces mitochondrial function to reduce hepatic glucose production and lipid metabolism [176,177]. In male mice on a 47% fructose diet for ten weeks, metformin increased the expression of genes regulating mitochondria biogenesis and antioxidants and reduced genes associated with DNL [178]. Metformin also prevents LPS-induced M1 macrophage polarization by activating AMPK reducing ROS production [179,180,181]. Taken together, these data suggest that metformin protects the liver from the onset of fructose-induced NAFLD by altering intestinal permeability and activation of hepatic Kupffer cells. 

Glucagon-like-peptide-1-receptor agonists (GLP-1-RA) are another class of drugs developed to treat type 2 diabetes. GLP-1, the peptide hormone that binds the GLP-1 receptor, is secreted by the L cells of the jejunum and ileum [182]. Fatty acids and sucrose are known to increase GLP-1 secretion, and GLP-1 causes the postprandial secretion of insulin in the pancreas [183,184]. In adult males, acute sucrose consumption causes a smaller increase in circulating GLP-1 levels compared with females [185]. The GLP-1 receptor is found in the kidneys, lungs, and hypothalamus [183]. Boland et al. demonstrated that the GLP-1 receptors also are expressed on mouse and human monocytes and macrophages [186]. GLP-1 decreases hunger and food intake [187,188,189]. GLP-1-RA suppresses appetite through AMPK inhibition in the hypothalamus [187,188]. Burmeister et al. demonstrated that intracerebroventricular injections of fructose attenuated the anorectic response induced by the GLP-1-RA, Exendin-4 (Ex-4) [188]. Fructose can activate AMPK and block the decrease in appetite and satiety associated with GLP-1 [190]. Likewise, Ghidewon et al. demonstrated that GLP-1-RA semaglutide decreased the intake of sucrose-enriched pellets in male mice [191]. GLP-1-RA combats metabolic disease through several mechanisms. For example, these drugs also increase the expression of GLUT4 receptors in adipose and muscle tissue, mediate lipid metabolism, and reduce the levels of ROS, leading to inflammation [192]. GLP-1-RA also increase glucose-dependent insulin secretion by pancreatic β-cells [182].

GLP-1-RAs have shown significant promise in treating NASH. In male rats administered fructose, GLP-1-RA significantly reduced hepatic steatosis by increasing b-catenin nuclear translocation, leading to a reduction in lipid synthesis enzymes, ACC, FAS, and stearoyl-coenzyme A desaturase-1 (SCD-1) [193]. Liraglutide, a GLP-1RA, reduced serum AST and ALT levels in hepatic ischemia–reperfusion injury [194]. GLP-1 receptor signaling also prevents M1 macrophage polarization [194]. The GLP-1 and glucagon receptor dual agonist cotadutide significantly reduced hepatic steatosis, inflammation, and fibrosis in NASH male mice on a high fructose, trans-fat, and cholesterol diet [195]. Ultimately, more research is needed to elucidate the effects of GLP-1-RAs on fructose absorption and metabolism in metabolic diseases. However, these drugs are still promising therapeutic targets for fructose-associated NAFLD.

## 4. Limitations and Future Directions

It is important to note that targeting fructose metabolism alone may not be sufficient for addressing the complexities of NAFLD, and a comprehensive approach involving dietary modifications, lifestyle changes, and management of metabolic risk factors is required. Notably, current research on fructose-induced disease progression and cellular metabolism is limited by the exaggerated concentrations of fructose used in studies. Depending on the location, specific cells and tissues would not be exposed to such high concentrations of circulating fructose. Interestingly, a high fructose consumption may be protective during acute liver injury. In male mice, fructose reduces acetaminophen-induced liver injury by increasing ChREBP and serum levels of fibroblast growth factor 21 [196]. Although we gain insight into fructose metabolism within these models, the current gap in our understanding regarding dietitian-recommended levels of fructose remains large. It is also necessary to note that human studies have demonstrated no correlation between fructose consumption and NAFLD development. Among older Finnish older, a high fructose intake was associated with a lower risk of NAFLD [197]. Similarly, total fructose intake was not different between patients with steatosis, NASH, or cirrhosis. Both studies were dependent on food frequency questionnaires and dietary recalls [198], which may lead to underreporting dietary intake of fructose. Thus, there is a need for metabolic tracing studies and biomarkers to track fructose consumption. 

In addition, very few groups have identified the effects of fructose on immune cell populations other than the plasticity of fructose-induced macrophage and B cell populations. Collectively, there are differences in fructose metabolism depending on tissue and cell type, mainly through the actions of KHK or HK. The availability and concentrations of fructose and glucose limit the initial steps in fructose metabolism. For example, residential intestine and liver cells utilize KHK-mediated metabolism, which shuttles primarily through glycolysis conversion intermediates. However, bone marrow-derived macrophages depend on oxidative phosphorylation and the TCA cycle to generate intermediates for cytokine secretion. Because of the differences in metabolism, it is unsurprising that fructose impacts parenchymal and nonparenchymal cellular function through different mechanisms. As research on fructose exposure and immune cells in disease increases, we may find novel metabolic pathways. 

The impact of sex on fructose metabolism and metabolic outcomes remains limited. Sex differences are reported for the effects of fructose metabolism on kidney function with fructose increasing GLUT5 protein expression in males compared with females [199]. Isotope tracing of fructose in healthy young males and females demonstrated that fructose is metabolized less efficiently in females, leading to reduced plasma insulin, lactate, and uric acid levels [200]. In animal models, fructose supplementation in the drinking water increased the body weight of male rats compared with female rats [201]. In addition, hepatic TNFα levels are reduced in female rats compared with male rats at 10 weeks of high-fructose feeding [202]. From the United Kingdom Biobank, fructose consumption from sugary drinks is associated with a 2% higher risk of hyperandrogenism among middle-aged women [130]. These findings suggest menopause or age in women alters fructose metabolism, leading to a possibly higher risk of metabolic diseases such as NAFLD. Elevations in androgens leads to increased plasma lipids, hepatic steatosis, and insulin resistance due to mitochondrial dysfunction in female rats [203]. Likewise, supplementation of 30% fructose in a high-fat diet increases liver weight, inflammation, and injury compared with high-fat-diet-fed ovariectomized female mice [204]. In males, fructose consumption causes metabolic disease but reduces testosterone and luteinizing hormone release in the anterior pituitary [205]. Studies linking mechanisms by which fructose regulates sex hormones and alters tissue specific fructose metabolism in males and females remain to be investigated.

## 5. Conclusions

NAFLD remains a significant health concern and managing its effects on hepatic lipid accumulation and insulin resistance is critical. To treat this condition effectively and prevent its progression to severe liver disease, we need a comprehensive approach that combines dietary modification, weight management, and pharmacological interventions. The literature discussed in this review defines the cellular and molecular mechanisms regulating fructose metabolism and the development of NAFLD. Fructose regulates lipid metabolism, inflammation, and immune cell function in the intestine, adipose tissue, and liver. Several studies show varying results depending on the concentrations of fructose supplied and the duration of the animal and clinical studies. These inconsistencies raise questions about how fructose affects individual cells, tissues, and biological systems. For example, how does fructose control subsets and functions of T cells, and how does it affect NAFLD? Also, understanding how fructose modulates intestinal function and identifying critical cellular and molecular signaling pathways are essential. Therefore, more research is needed to understand the critical pathways regulated by fructose in metabolic systems and the pharmacological interventions targeting various steps in fructose metabolism. By addressing fructose metabolism, we can effectively manage NAFLD and prevent its progression to more severe liver disease. 

## Figures and Tables

**Figure 1 biomolecules-14-00845-f001:**
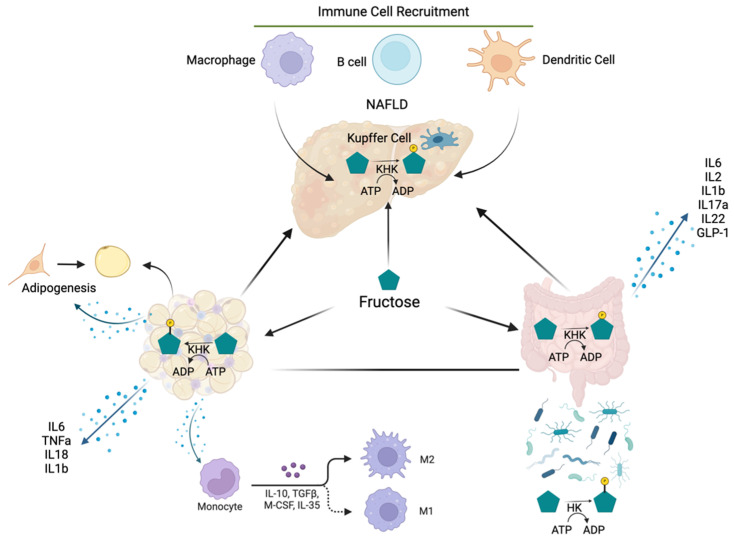
Direct and indirect effects of fructose in metabolic tissues. Fructose exposure can lead to recruitment of immune cells to metabolic tissues such as intestine, liver, and adipose. This recruitment leads to increased secretion of pro-inflammatory cytokines, aiding in cell-to-cell signaling. Fructose activates adipogenesis via upregulation of proteins involved in differentiation, monocyte and macrophage polarization, and cytokine secretion.

**Figure 2 biomolecules-14-00845-f002:**
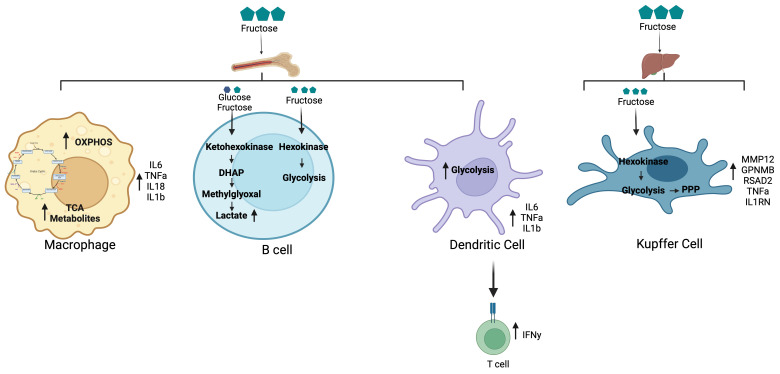
Fructose metabolism and immune cells. Bone marrow-derived immune cells, such as macrophages, B cells, and dendritic cells, display a pro-inflammatory phenotype when exposed to fructose; however, the mechanism for this response is unique for each cell type. Macrophages rely on oxidative phosphorylation and TCA cycle passes for energy and cytokine production. Interestingly, B cells and dendritic cells utilize intermediates of glycolysis conversion to keep up with the energy demands of the cell. Secretion of the cytokines IL-6, TNFα, IL-18, and IL-1β aid in cell-to-cell communication as well as priming an immune response. This signaling can lead to recruitment of immune cells within metabolic tissues such as liver, intestine, and adipose, thus increasing overall inflammation. Cytokine secretion can directly activate T cells through TNFα inducing secretion of IFNγ. Kupffer cells depend on glycolysis and the pentose phosphate pathway for regulation of genes associated with wound healing and anti-inflammatory response such as MMP12, GPNMB, RSAD2, TNFα, and IL1RN.

**Figure 3 biomolecules-14-00845-f003:**
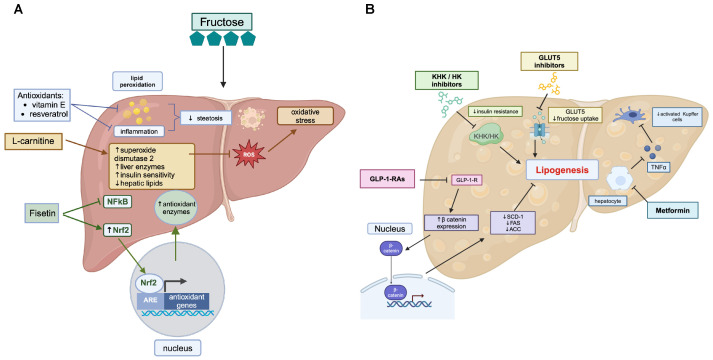
The hepatic effects of nutritional supplements and pharmacological interventions in fructose-induced NAFLD. (**A**) The hepatic effects of nutritional supplements in fructose-induced NAFLD. Antioxidant supplements, such as vitamin E and resveratrol, are shown to alleviate fructose-induced hepatic steatosis and liver damage through the attenuation of lipid peroxidation and inflammation. L-carnitine also facilitates antioxidant properties through increased expression of superoxide dismutase 2 (SOD2), which catalyzes the dismutation of ROS, reducing oxidative stress. L-carnitine also is suggested to increase levels of liver enzymes and insulin sensitivity in hepatocytes and alleviate hepatic lipid accumulation. Additionally, the flavonoid fisetin is suggested to alleviate fructose-induced liver damage and inflammation through suppressing NFkB. Fisetin also activates Nrf2, a transcription factor that binds the antioxidant response element (ARE), a DNA sequence found in the promoter regions of several genes encoding antioxidant enzymes, increasing the expression in the liver. (**B**) The hepatic effects of pharmacological interventions in fructose-induced NAFLD. Several pharmacological interventions target fructose-induced hepatic DNL. GLUT5 inhibitors alleviate fructose-induced DNL through decreasing fructose uptake and metabolism, resulting in increased insulin sensitivity. Likewise, ketohexokinase (KHK) and hexokinase (HK) inhibitors decrease fructose-induced DNL by reducing hepatic fructose metabolism, improving insulin sensitivity. Similarly, GLP-1 receptor agonists (GLP-1-RAs) function to decrease hepatic DNL by increasing gene expression of the transcription factor β-catenin, which translocates to the nucleus and mediates the downregulation of acetyl-CoA carboxylase (ACC), fatty acid synthase (FAS), and stearoyl-CoA desaturase-1 (SCD-1). Apart from its effects in increasing insulin sensitivity, metformin plays a protective role against NAFLD by reducing hepatic expression of TNFα and inhibiting the overactivation of hepatic Kupffer cells, reducing inflammation.

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
