# Peer review of "Regulation of Fructose Metabolism in Nonalcoholic Fatty Liver Disease"

_biomolecules, 2024, doi:10.3390/biom14070845_

Round 1
Reviewer 1 Report (Previous Reviewer 2)
Comments and Suggestions for Authors
Dear Authors;
Thank you very much for the corrections you made. The manuscript is good, and I would like to recommend it for publication in the journal.
Kind regards,
Comments on the Quality of English LanguageMinor corrections are required.
Author Response
We have had an editor read over the manuscript for edits and made corrections.
Reviewer 2 Report (Previous Reviewer 1)
Comments and Suggestions for Authors
This is a comprehensive review article with an extensive assessment of published studies. Given the current scenario of changing disease nomeclatures with its inherent burden of confusion, controversies and acrimony (PMID: 36950481; Byrne, CD.; Targher G. MASLD, MAFLD, or NAFLD criteria: have we re-created the confusion and acrimony surrounding metabolic syndrome?. Metab Target Organ Damage. 2024, 4, 10. http://dx.doi.org/10.20517/mtod.2024.06) authors may be willing to explain the reason(s) why they have adopted the old NAFLD nomenclature as opposed to MAFLD and MASLD. Please, explain whethe this is due to to the desire to maintain alignment with published studies or to other reasons resulting from their investigators' responsibilities (PMID: 38336589). In this regard, please cite any studies that support or reject the notion that NAFLD and MASLD identify the same patient population.
Additionally, a novel possible utility of fructose as a novel nutraceutical for Acetamine -induced liver injury should be discussed (Rodrigues, K.; Hussain R.; Cooke S.; Zhang G.; Zhang D.; Yin L.; Tong X. Fructose as a novel nutraceutical for acetaminophen (APAP)-induced hepatotoxicity. Metab Target Organ Damage. 2023, 3, 20. http://dx.doi.org/10.20517/mtod.2023.28).
Author Response
We have selected NAFLD based on the current publications referenced in the manuscript using the terminology NAFLD instead of MAFLD.
To the limitations section we added fructose as possible nutraceutical for APAP hepatotoxicity.
This manuscript is a resubmission of an earlier submission. The following is a list of the peer review reports and author responses from that submission.
Round 1
Reviewer 1 Report
Comments and Suggestions for Authors
GENERAL COMMENT
The evolving scenario of rapidly changing nomenclatures in the arena of NAFLD/NASH, and MAFLD/MASLD has admittedly generated some confusion among practitioners and investigators (PMID: 36950481). There are several studies highlighting similarities and differences between NAFLD and MAFLD (see, for example doi: 10.14218/JCTH.2022.00154. doi: 10.3350/cmh.2022.0367. PMID: 36179795 PMID: 32930521 and others) and between NAFLD and MASLD (see, for example PMID: 38286339, PMID: 38293788 PMID: 38224780, PMID: 38135250, PMID: 38112428) and others) or between MAFLD and MASLD (PMID: 38127259). On these grounds, authors may consider renaming “MAFLD/MASH” to “NAFLD/NASH” throughout their manuscript. Alternatively, all above references should be changed.
Moreover, the manuscript should also include additional sections on epidemiological evidence associating fructose consumption with the risk of (prevalent/incident) NAFLD, sex differences in fructose metabolism, cardiovascular risk associated with steatotic liver disease and fructose consumption, practical suggestions to NAFLD patients as to certain lifestyle habits (e.g. diet and physical activity) with specific regard to their impact on fructose metabolism. Finally, the manuscipt appears to be insufficiently referenced and, to address this limitations, bibliography recommended below should also be discussed.
SPECIFIC COMMENT
MAJOR
1. Lines 33-35 must be reworked to better transmit the notion that, collectively, the “metabolic fatty livers syndromes (NAFLD/NASH, and MAFLD/MASLD), irrespective of how we want to call them, are a GLOBAL issue, as opposed to a regional concern (doi: 10.1007/s12072-023-10568-z.)
2. Lines 35-37 refer to NAFLD (not MAFLD). Steatosis must be defined.
3. Lines 38-39 refer to NASH (not MASH).
4. Line 41: references 9, and 11-16 all refer to NAFLD (not to MASH).
5. Line 47: References 17 to 20 refer to NAFLD (not MAFLD).
6. Line 49: References 21-23 allude to NAFLD (not MAFLD). Additionally, the old notion of “NAFLD as a manifestation of the metabolic syndrome” fails to render the notion that the relationship is mutual and bi-directional (PMID: 32824337).
7. Line 505 “Discussion and Future Direction”. Rephrase to “Research agenda and conclusions” and, if possible, split it into two different sections: “Research agenda” and “Conclusion”.
8. NAFLD has definite sexual dimorphism (doi: 10.1016/j.cgh.2020.04.067, PMID: 32354182, PMID: 33173188). Is there any evidence that fructose metabolism is different in men and women? For example, a recent study highlighted that Fructose intake from sugar-sweetened beverages was associated with a greater risk of hyperandrogenism in women (PMID: 38291515). Therefore, while appreciating the discussion of organ-specific effect of fructose, I recommend extensively discussing also the less appreciated and more subtle sex-specific aspects of fructose metabolism (PMID: 30883738, PMID: 30205493).
9. Authors may be willing to articulate an exhaustive discussion of the epidemiological evidence supporting the notion that increased consumption of fructose is indeed associated with a parallel increased risk of NAFLD risk.
10. Dietary intakes of fructose (and vitamin C) are associated with prevalent hyperuricemia in community-based studies (PMID: 29546301) suggesting that this may be a specific biochemical pathway leading to increased risk of NAFLD which, in its turn, is also associated with increased SUA levels (PMID: 37305378).
11. Discuss the important notion that increased physical activity blunts the fructose-induced glycemic response among healthy subjects (PMID: 24848627) given that this does have clinical implications.
12. Authors may be willing to discuss whether fructose contained in fruit is equally “toxic” for liver’s health as industrially added fructose (e.g. sweetened beverages and other foodstuff). Is this a matter of different concentrations? What dietary suggestions should be given to NAFLD patients regarding consumption of fresh fruit? (http://dx.doi.org/10.20517/mtod.2021.11).
13. NAFLD/MAFLD/MASLD is currently conceptualized as a major cardiovascular risk factor irrespective of confounding factors (doi: 10.1038/s41575-023-00880-2). Therefore, a specific section of the manuscript should be dedicated to discussing the impact of fructose on cardiovascular risk (PMID: 28548281 PMID: 32916151, PMID: 32599713 PMID: 26358358 PMID: 26178027 PMID: 31885782).
14. Based on scientific evidence, do authors believe that a specific tax system should be used to discourage excess fructose consumption ? I think this would be a valuable contribution for the public if deliverd by independent scientists as opposed to policy-makers.
MINOR
1. Abstract, line 26 - I am not happy with this sentence: “Therefore, we will better understand”, please rephrase. Alternatives such as “these findings are critically discussed” or similar may be considered.
2. Line 174 “in mammals and humans” I think that “mammals” also includes “humans” in most cases.
3. Should comment on a study going against the current epidemiological paradigms (PMID: 25099548). Are there any methodological issues explaining the unexpected findings ?
4. Regarding the so-called “secondary NAFLD forms” please, discuss also PMID: 36090199.
Author Response
We thank the reviewers for the critical review and feedback. We have made considerable changes to the manuscript and highlighted the changes in red in the manuscript.
Reviewer 1
GENERAL COMMENT
The evolving scenario of rapidly changing nomenclatures in the arena of NAFLD/NASH, and MAFLD/MASLD has admittedly generated some confusion among practitioners and investigators (PMID: 36950481). There are several studies highlighting similarities and differences between NAFLD and MAFLD (see, for example doi: 10.14218/JCTH.2022.00154. doi: 10.3350/cmh.2022.0367. PMID: 36179795 PMID: 32930521 and others) and between NAFLD and MASLD (see, for example PMID: 38286339, PMID: 38293788 PMID: 38224780, PMID: 38135250, PMID: 38112428) and others) or between MAFLD and MASLD (PMID: 38127259). On these grounds, authors may consider renaming “MAFLD/MASH” to “NAFLD/NASH” throughout their manuscript. Alternatively, all above references should be changed.
Moreover, the manuscript should also include additional sections on epidemiological evidence associating fructose consumption with the risk of (prevalent/incident) NAFLD, sex differences in fructose metabolism, cardiovascular risk associated with steatotic liver disease and fructose consumption, practical suggestions to NAFLD patients as to certain lifestyle habits (e.g. diet and physical activity) with specific regard to their impact on fructose metabolism. Finally, the manuscript appears to be insufficiently referenced and, to address these limitations, bibliography recommended below should also be discussed.
SPECIFIC COMMENT
MAJOR
- Lines 33-35 must be reworked to better transmit the notion that, collectively, the “metabolic fatty livers syndromes (NAFLD/NASH, and MAFLD/MASLD), irrespective of how we want to call them, are a GLOBAL issue, as opposed to a regional concern (doi: 10.1007/s12072-023-10568-z.)
Thank you to the reviewer, we have switched the terminology back to NAFLD/NASH throughout the document based on the current and past literature referenced in the review article. We have added the reference to highlight the global impact of NAFLD/NASH.
- Lines 35-37 refer to NAFLD (not MAFLD). Steatosis must be defined.
We have defined steatosis and refer to NAFLD.
- Lines 38-39 refer to NASH (not MASH).
Please see Response to Question 1.
- Line 41: references 9, and 11-16 all refer to NAFLD (not to MASH).
Please see Response to Question 1.
- Line 47: References 17 to 20 refer to NAFLD (not MAFLD).
Please see Response to Question 1.
- Line 49: References 21-23 allude to NAFLD (not MAFLD). Additionally, the old notion of “NAFLD as a manifestation of the metabolic syndrome” fails to render the notion that the relationship is mutual and bi-directional (PMID: 32824337).
Please see Response to Question 1.
- Line 505 “Discussion and Future Direction”. Rephrase to “Research agenda and conclusions” and, if possible, split it into two different sections: “Research agenda” and “Conclusion”.
As suggested by the reviewer, we have rephrased the section to “Limitations and Future Directions” and “Conclusion”.
- NAFLD has definite sexual dimorphism (doi: 10.1016/j.cgh.2020.04.067, PMID: 32354182, PMID: 33173188). Is there any evidence that fructose metabolism is different in men and women? For example, a recent study highlighted that Fructose intake from sugar-sweetened beverages was associated with a greater risk of hyperandrogenism in women (PMID: 38291515). Therefore, while appreciating the discussion of organ-specific effect of fructose, I recommend extensively discussing also the less appreciated and more subtle sex-specific aspects of fructose metabolism (PMID: 30883738, PMID: 30205493).
We agree with the reviewer and distinguished fructose studies conducted in men and women or male and female animal models in the manuscript. We have also highlighted sex differences in the limitation of the studies conducted on fructose metabolism and metabolic outcomes. We did not provide an extensive discussion of the hyperandrogenism in women because the findings relating to NAFLD and fructose are limited. We have added the current research findings on sex differences, fructose metabolism and hyperandrogenism to the limitations section of the manuscript Lines 533-551.
- Authors may be willing to articulate an exhaustive discussion of the epidemiological evidence supporting the notion that increased consumption of fructose is indeed associated with a parallel increased risk of NAFLD risk.
Thank you for the suggestion. We have included recent manuscripts on the epidemiological evidence supporting increased consumption of fructose and the risk of NAFLD Lines 67-70. We did not include an exhaustive discussion because the major goal for the article is to discuss the molecular mechanisms by which fructose drives NAFLD.
- Dietary intakes of fructose (and vitamin C) are associated with prevalent hyperuricemia in community-based studies (PMID: 29546301) suggesting that this may be a specific biochemical pathway leading to increased risk of NAFLD which, in its turn, is also associated with increased SUA levels (PMID: 37305378).
Thank you for the information. We have included a paragraph included the suggested references Lines 67-69.
- Discuss the important notion that increased physical activity blunts the fructose-induced glycemic response among healthy subjects (PMID: 24848627) given that this does have clinical implications.
Thank you for the reference we have added the manuscript to the section on dietary modifications Lines 355-357.
- Authors may be willing to discuss whether fructose contained in fruit is equally “toxic” for liver’s health as industrially added fructose (e.g. sweetened beverages and other foodstuff). Is this a matter of different concentrations? What dietary suggestions should be given to NAFLD patients regarding consumption of fresh fruit? (http://dx.doi.org/10.20517/mtod.2021.11).
Fruit juice versus fruit consumption is reported to have a greater association with NAFLD development in patients. Fruit contains other antioxidants found in the skin that have anti-inflammatory properties. In addition, depending on the type of fruit, the content of fiber may help to reduce fructose absorption and metabolism in the intestine. We also believe this topic would be suited for an epidemiological or nutrition-based manuscript to make recommendations about the toxicity of fructose from fruit and recommendations for NAFLD patients. Chen et al. reported fructose consumption from sweetened beverages increased the risk of hyperandrogenism in women. Buziau et al. 2022 (https://doi.org/10.2337/dc21-2123) and Li et al. 2023 (https://doi.org/10.1016/j.ajcnut.2023.01.006) also reported fructose consumption from sweetened beverages increases the risk of cardiometabolic disorders and hepatic steatosis in adults.
NAFLD/MAFLD/MASLD is currently conceptualized as a major cardiovascular risk factor irrespective of confounding factors (doi: 10.1038/s41575-023-00880-2). Therefore, a specific section of the manuscript should be dedicated to discussing the impact of fructose on cardiovascular risk (PMID: 28548281 PMID: 32916151, PMID: 32599713 PMID: 26358358 PMID: 26178027 PMID: 31885782).
We thank the reviewer for the recommendation and agree based on the literature NAFLD cardiovascular disease is a strong risk factor for NAFLD. We have included the references in the manuscript Lines 68-69. However, in this manuscript we are focusing on tissues that can drive NAFLD progression versus cardiovascular risk. This is a secondary response of NAFLD and focused more on how fructose drives NAFLD progression via absorption in the intestine and or increased adiposity which leads to increased inflammation which can drive liver dysfunction.
- Based on scientific evidence, do authors believe that a specific tax system should be used to discourage excess fructose consumption? I think this would be a valuable contribution for the public if delivered by independent scientists as opposed to policy-makers.
We agree the tax system is a possible solution to excess fructose consumption based dietary reductions of fructose reduce fructose metabolism and fatty liver disease in mice and humans. However, because the focus of the manuscript is on molecular metabolic signaling and cellular responses of fructose on metabolic tissues and immune cells, we decided to refrain from including a discussion in the review on taxing high sugar beverages and foods. To make this statement we would need to do an extensive review of the literature regarding countries that implemented taxes and the impact on NAFLD. Currently there are limited studies correlating increasing taxes on sugary beverages and NAFLD, thus we are unable to make a strong statement and decided to not include statement in the review.
MINOR
- Abstract, line 26 - I am not happy with this sentence: “Therefore, we will better understand”, please rephrase. Alternatives such as “these findings are critically discussed” or similar may be considered.
Thank you. We changed the statement as requested.
- Line 174 “in mammals and humans” I think that “mammals” also includes “humans” in most cases.
We removed mammals and clarified the species human, rats, mice, etc.
- Should comment on a study going against the current epidemiological paradigms (PMID: 25099548). Are there any methodological issues explaining the unexpected findings?
We included this study in the limitations section (Lines 513-516) and highlight the need for direct assessment of fructose consumption. This study was completed in only Finnish adults, where lifestyle choices may create biases such as Europeans inducing more physical activity than adults in the United States for example. The population is different compared to the United States. Dietary intake of fructose and other foods was assessed via dietary questionnaires. Responses could have been embellished/unreflective of the participants actual diet due to self-report bias. Therefore, direct measurement or tracking of fructose that is quantifiable is required.
- Regarding the so-called “secondary NAFLD forms” please, discuss also PMID: 36090199.
We have changed MALFD to NAFLD and MASH to NASH throughout the manuscript.
Reviewer 2 Report
Comments and Suggestions for Authors
Dear Authors;
The aim of the manuscript was to describe the regulation of fructose metabolism in metabolic dysfunction-associated fatty liver disease. However, I suggest reviewing the document because several instances of plagiarism have been found. You can find the information in the attached file.
In relation to the manuscript:
1. A table summarizing the main findings related to fructose metabolism in metabolic dysfunction-associated fatty liver disease should be added.
2. A figure explaining the effects of nutritional supplements on MAFLD should be added.
3. A figure explaining the effects of pharmacological interventions on MAFLD should be added.
4. Limitations must be added.
Kind regards,

Author Response
We thank the reviewers for the critical review and feedback. We have made considerable changes to the manuscript and highlighted the changes in red in the manuscript.
Reviewer 2
Dear Authors;
The aim of the manuscript was to describe the regulation of fructose metabolism in metabolic dysfunction-associated fatty liver disease. However, I suggest reviewing the document because several instances of plagiarism have been found. You can find the information in the attached file.
We analyzed the information attached regarding plagiarism but not see the sentences that were identified. We used Grammarly to check for plagiarism and found no incidence of plagiarism. We have updated all sections that were missing references.
In relation to the manuscript:
- A table summarizing the main findings related to fructose metabolism in metabolic dysfunction-associated fatty liver disease should be added.
We thank the reviewer for the suggestion and have instead included figure to demonstrating how fructose metabolism regulates NAFLD. The figure highlights the cell types impacted, how the cells metabolism of fructose, and the impact on the progression of NAFLD.
- A figure explaining the effects of nutritional supplements on MAFLD should be added.
We have included a figure highlighting the nutritional supplements on NAFLD.
- A figure explaining the effects of pharmacological interventions on MAFLD should be added.
We have included a figure highlighting the pharmacological interventions on NAFLD.
- Limitations must be added.
We included a section on limitations highlighting length of studies, concentration of fructose, and sex differences.
Reviewer 3 Report
Comments and Suggestions for Authors
This is an interesting review considering the consequences of fructose exposure and potential for initiating/exacerbating MASLD. It is generally clearly written although there is a tendency to overstatement in places and evidence of unsubstantiated statements where reference citations should be added. The abstract suggests that impact on fibrosis has been considered – it hasn’t and this statement should be removed.
Early in the intro section the authors cite metrics for MASLD indicence and progression rates. These are based on what are now outdated references and it would be useful to reconsider newer data to be sure HCC rate in patients with MAFLD is not overstated. Similarly the discussion around the landscape of drug trials should probably include acknowledgement that the FDA has fast track approved the Pfizer drug combination for NASH and statements around the Phase 3 trials of
resmetirom.
The immunological consequences of Fructose metabolism are also perhaps a little naively considered with an emphasis on myeloid cells and B cells in the main. Given the role of other myeloid cells such as neutrophils and specific lymphocyte subpopulations in disease initiation and progression one might also expect some narrative including these cells. The macrophage liver section strays into discussion of phenotype and origins rather than impact of fructose on these. Need to cite evidence for hepatic differentiation or altered recruitment after fructose exposure specifically (ie not due to other drivers of MASLD) to fit with the scope of the article.
The authors should comment on the limitations of in vitro/vivo exposure studies when – many use dramatically high concentrations well above exposure levels in tissue/peripheral circulation, so conslusions should be tempered accordingly.
The section on human trials at the end is nice but the authors could expand on the information detailing human trials of KHK inhibitors to discuss MOA and likely drivers of safety signals. This would allow the reading to consider likely future for future more targetted compounds impacting on fructose metabolic pathways.
Minor comments:
Needs a good proofread for typos, spacing and grammatical errors. Some unattributed statements require citations eg section around lines 174-189
Also thorough check for citation errors eg "Mice on a 20% fructose diet for 9 weeks showed increased hepatic fatty acid accumu lation through upregulated CD36 signaling [26]." line 246. Think this is not correct reference
Typo line 459 "muscle (ref). Inhibiting"
Comments on the Quality of English Language
Minor grammatical and proofreading errors thoughout need attention.
Author Response
This is an interesting review considering the consequences of fructose exposure and potential for initiating/exacerbating MASLD. It is generally clearly written although there is a tendency to overstatement in places and evidence of unsubstantiated statements where reference citations should be added. The abstract suggests that impact on fibrosis has been considered – it hasn’t and this statement should be removed.
We have changed the use of MASLD throughout the document and used the NAFLD terminology due to the majority of articles referenced based on NAFLD. We agree with the reviewer and removed fibrosis in the manuscript.
Early in the intro section the authors cite metrics for MASLD incidence and progression rates. These are based on what are now outdated references and it would be useful to reconsider newer data to be sure HCC rate in patients with MAFLD is not overstated. Similarly, the discussion around the landscape of drug trials should probably include acknowledgement that the FDA has fast track approved the Pfizer drug combination for NASH and statements around the Phase 3 trials of resmetirom.
Thank you for the critique. We have updated the rate of NAFLD to the most recent data findings. We have also included resmetirom in the introduction as the only approved NASH therapy. Lines 31-55
The immunological consequences of Fructose metabolism are also perhaps a little naively considered with an emphasis on myeloid cells and B cells in the main. Given the role of other myeloid cells such as neutrophils and specific lymphocyte subpopulations in disease initiation and progression one might also expect some narrative including these cells. The macrophage liver section strays into discussion of phenotype and origins rather than impact of fructose on these. Need to cite evidence for hepatic differentiation or altered recruitment after fructose exposure specifically (ie not due to other drivers of MASLD) to fit with the scope of the article.
We agree with the reviewer and have modified the section discussing the myeloid cell populations. Currently there is no data on neutrophils and other lymphocyte populations and fructose metabolism and phenotype. We have also referenced our research groups manuscript published in Scientific Reports that demonstrates how fructose regulates hepatic immune cell populations. In this study a 30% fructose diet increases transitioning monocytes that differentiate into hepatic macrophages. Interestingly, we discovered fructose reduces Kupffer cells and B cells. Lines 159-162, 253-258, and 320-336.
The authors should comment on the limitations of in vitro/vivo exposure studies when – many use dramatically high concentrations well above exposure levels in tissue/peripheral circulation, so conslusions should be tempered accordingly.
Thank you for the suggestion. We included in the limitations section the high concentrations used in many of the in vitro and animal studies at the end of the review article.
The section on human trials at the end is nice but the authors could expand on the information detailing human trials of KHK inhibitors to discuss MOA and likely drivers of safety signals. This would allow the reading to consider likely future for future more targeted compounds impacting on fructose metabolic pathways.
We have included safety signals and MOA in the section on KHK inhibitors Lines 401-412.
Minor comments:
Needs a good proofread for typos, spacing and grammatical errors. Some unattributed statements require citations eg section around lines 174-189
Thank you for the feedback. We used an outside source to proofread the manuscript and made the suggested changes to correct the spacing and grammatical errors. We condensed the section on fructose metabolism and added the needed references.
Also thorough check for citation errors eg "Mice on a 20% fructose diet for 9 weeks showed increased hepatic fatty acid accumulation through upregulated CD36 signaling [26]." line 246. Think this is not correct reference
Thank you for identifying this error. We have removed the sentence and reference.
Typo line 459 "muscle (ref). Inhibiting"
We corrected the typo.
Minor grammatical and proofreading errors thoughout need attention.
Thank you for the comment. We had an outside company proofread the document and made the suggested corrections.
Reviewer 4 Report
Comments and Suggestions for Authors
General Comment
According to the Introduction (ln 79-81), in this review the authors’ goal is to “highlight the current literature on how fructose regulates inflammation in metabolic tissues, the implications of fructose metabolism on immune cell phenotypes, and therapeutic targets of fructose metabolism”. However, most of the text is dedicated to fructose metabolism, which is extensively reviewed (sections 2.1 to 2.5, ln 85 to 297). In contrast, only two paragraphs (2.5.1 and 2.5.2, ln 298-327) deal with the impact of fructose on immune cells.
In my opinion, the review would gain in readability and interest if the authors shorten the paragraphs related to fructose metabolism and focus on novel and scientifically sound contributions regarding the regulation of immune cell function by fructose.
Major comments
- Figures are not referenced in the text
- Figure 1 depicts some mechanisms that only appear in the legend, but not in the article. Specifically, the figure refers to the activation of adipogenesis through the induction of extracellular vesicle release and stem cells stimulation, and reduction of macrophage activation also via EV release. All these aspects should be covered in the review, with the appropriate references.
- From ln 157 to 188 there are no references, which must be added. References must be added also to identify the studies cited in ln 402, 411 and 459.
- Ln 396 to 414: the effects of inhibiting KHK is discussed, highlighting the effects of specific enzyme inhibitors, some of them with important safety concerns. Recently, it was shown that bempedoic acid, an approved drug with a good safety profile, inhibits KHK and reduces hepatic steatosis in a rat model of MAFLD based on the administration of a high-fat diet and 10% fructose in drinking water. This reference should be added to this paragraph (Velázquez et al. Biomedicines 2022;10(7):1517).
- Ln 438-440: how does the inhibition of hepatic GLUT5 impacts the metabolism of fructose in the intestine e and adipose tissue? By increasing substrate availability in non-hepatic tissues?
- Ln 461: I don’t agree in that these findings “validate” the therapeutic potential of GLUT5 inhibitors
- Ln 488-494: this paragraph is difficult to understand it should be re-written
- In 499: cotadutide is a GLP-1 and glucagon receptor dual agonist.
- ln 501, reference 126 is a review. The original study showing the effects of cotadutide must be cited (Boland et al, Nat Metab 2, 413-431, 2020).
Minor comments
- There are some blank spaces in the text, such as ln 47, 142, 352, 456
- Ln 67: Fruits do not contain HFCS, as it seems to be inferred from the sentence
- Ln 116: ChREBP, not chREBP
- ln 140: define TCA (first cited)
- ln 239: acetyl-CoA, not acetyl-coA
- ln 469: define RBP-4
- ln 472: by and increasing (remove and)
- ln 508: distinct differences
- ln 516: effects or affects?
Comments on the Quality of English LanguageSome minor corrections needed (see comments to the author)